# Pharmacokinetics and Metabolomic Profiling of Metformin and *Andrographis paniculata*: A Protocol for a Crossover Randomised Controlled Trial

**DOI:** 10.3390/jcm11143931

**Published:** 2022-07-06

**Authors:** Khim Boon Tee, Luqman Ibrahim, Najihah Mohd Hashim, Mohd Zuwairi Saiman, Zaril Harza Zakaria, Hasniza Zaman Huri

**Affiliations:** 1Department of Clinical Pharmacy and Pharmacy Practice, Faculty of Pharmacy, Universiti Malaya, Kuala Lumpur 50603, Malaysia; teekhimboon12@gmail.com; 2National Pharmaceutical Regulatory Agency, Ministry of Health Malaysia, Petaling Jaya 46200, Malaysia; zaril@npra.gov.my; 3Department of Medicine, Faculty of Medicine, Universiti Malaya, Kuala Lumpur 50603, Malaysia; email@luqmanibrahim.com; 4Department of Pharmaceutical Chemistry, Faculty of Pharmacy, Universiti Malaya, Kuala Lumpur 50603, Malaysia; najihahmh@um.edu.my; 5Centre for Natural Products Research and Drug Discovery, Universiti Malaya, Kuala Lumpur 50603, Malaysia; zuwairi@um.edu.my; 6Institute of Biological Science, Faculty of Science, Universiti Malaya, Kuala Lumpur 50603, Malaysia; 7Centre for Research in Biotechnology for Agriculture (CEBAR), Universiti Malaya, Kuala Lumpur 50603, Malaysia; 8Clinical Investigation Centre, Universiti Malaya Medical Centre, Kuala Lumpur 50603, Malaysia

**Keywords:** pharmacokinetics, metabolomics, metformin, *Andrographis paniculata*, clinical trials

## Abstract

This protocol aims to profile the pharmacokinetics of metformin and *Andrographis paniculata* (AP) and continue with untargeted pharmacometabolomics analysis on pre-dose and post-dose samples to characterise the metabolomics profiling associated with the human metabolic pathways. This is a single-centre, open-labelled, three periods, crossover, randomised-controlled, single-dose oral administration pharmacokinetics and metabolomics trial of metformin 1000 mg (*n* = 18), AP 1000 mg (*n* = 18), or AP 2000 mg (*n* = 18) in healthy volunteers under the fasting condition. Subjects will be screened according to a list of inclusion and exclusion criteria. Investigational products will be administered according to the scheduled timeline. Vital signs and adverse events will be monitor periodically, and standardized meals will be provided to the subjects. Fifteen blood samples will be collected over 24 h, and four urine samples will be collected within a 12 h period. Onsite safety monitoring throughout the study and seven-day phone call safety follow-up will be compiled after the last dose of administration. The plasma samples will be analysed for the pharmacokinetics parameters to estimate the drug maximum plasma concentration. Untargeted metabolomic analysis between pre-dose and maximum plasma concentration (Cmax) samples will be performed for metabolomic profiling to identify the dysregulation of human metabolic pathways that link to the pharmacodynamics effects. The metformin arm will focus on the individualised Cmax plasma concentration for metabolomics study and used as a model drug. After this, an investigation of the dose-dependent effects will be performed between pre-dose samples and median Cmax concentration samples in the AP 1000 mg and AP 2000 mg arms for metabolomics study. The study protocol utilises a crossover study design to incorporate a metabolomics-based study into pharmacokinetics trial in the drug development program. The combination analyses will complement the interpretation of pharmacological effects according to the bioavailability of the drug.

## 1. Introduction

Phase one trials aim to estimate the pharmacokinetics and pharmacodynamics of the investigational products, measure drug activity, and estimate the initial safety and tolerability [1]. According to the clinicaltrials.gov database, it was found that 1.5% of early-phase clinical trials applied metabolomics investigation in the studies from 2004 to 2015 [2]. The application of metabolomics in drug discoveries includes the investigation of a drug’s mechanism of action, identify the drug targets and monitor efficacy outcomes [3]. Pharmacometabolomics could be used to identify relevant biomarkers and validate the complex methodology in the early drug development process. Metabolomics analyses the end-products of the cellular reactions, closely related to the phenotype [4]. In the discovery and development of drugs, the metabolomics approach enables diagnostic biomarker identification, drug target screening in absorption, disposition, metabolism, elimination and toxicity, and applications in clinical trials to monitor pharmacological as well as adverse effects [5].

Herbal medicine is a multi-components therapeutic that combines various compounds interacting with multiple targets to produce the optimum pharmacological effects [6]. Single biomarker monitoring is no longer sufficient to evaluate the therapeutic effects of high-dimensional treatment methods from herbal medicine. Integrating metabolomics and pharmacokinetics to study multi-component drugs has expanded the application of metabolomics in both drugs and herbal medicine [7]. Various studies have included metabolomics-based research in the pharmacokinetic studies of single chemicals and multi-components drugs [8]. The incorporation of metabolomics study into an early-phase pharmacokinetic study could simultaneously quantitate the rate and extent of drug absorption into the human blood circulation and explore the metabolic pathway involved in the early-phase clinical trial.

Metformin is a well-known first line antidiabetic therapeutic agent that have established safety profiles, the mechanism of action is inconclusive. The common side effects are generally related to gastrointestinal and the effects are reversible [9]. The pharmacokinetic parameters for metformin 1000 mg includes the maximum plasma concentration (Cmax) of 1560 ng/mL, the area under the plasma drug concentration time curve (AUC_0-infinity_) of 9360 ng*h/mL, and time to reach Cmax (Tmax) of 2.5 h [10]. AP is a herbal medicine from the Acanthaceae family found in Southern and Southeast Asia [11] with several pharmacological activities including antidiabetic, anti-inflammatory, antibacterial, antiviral, and others [12]. There are several A. paniculata products registered with the Ministry of Health Malaysia as traditional medicines without therapeutic claims. The major bioactive compounds for this multi-component medicinal plant includes andrographolide, deoxyandrographolide, neoandrographolide, 14-deoxy-11,12-didehydroandrographide, and isoandrographolide [13]. A meta-analysis of ten randomised controlled clinical trials demonstrated that AP is generally safe, and the common adverse events for AP are gastrointestinal disorder, skin, and subcutaneous disorder [14]. Several pharmacokinetic trials have been conducted using an AP of fixed combination for andrographolide [15,16,17]. One study investigated up to four major compounds for AP using multiple doses in healthy volunteers, the Cmax for andrographolide was 32.41 ng/mL, and the Cmax for 14-deoxy-11,12-didehydroandrographolide was 44.89 ng/mL [18]. A clinical trial conducted in Malaysia using AP capsules up to 1200 mg for 12 weeks significantly reduced HbA1c by 5.46% [19]. Due to insufficient clinical data associated with the pharmacokinetics of AP and its pharmacological effects, the registration of herbal medicine with therapeutic claims remains a challenge. The quality and consistency of major phytochemicals in herbal medicines affect the pharmacokinetic profiles. As a result, we performed a pharmacokinetics and metabolomics study in a similar group of healthy volunteers from the same batch of investigational products.

The phase one study aims to find an optimal dose through dose escalation in several cohorts [20]. This is a proof-of-concept study to explore the integration of the metabolomic study and the pharmacokinetic study in early-phase clinical trial. In the phase 1 trial, clinical monitoring, continuous laboratory parameters, and pharmacokinetics study are performed to determine dose-escalation. Metabolomic profiles obtained before, during, or after dosing provide insights about the drug mechanism of action and variation in the response to treatment [21]. The application of a metabolomic study in the early-phase clinical trial has the advantage of a small, short, and underpowered design, however, pharmacometabolomics can help reduce the variability of the study populations and act as a powerful surrogate of drug response in the identification of perturbed human metabolic pathways and explore provisional therapeutic or safety biomarkers [2].

In the bioanalytical analyses, metformin will be used as a model drug for individualised pharmacokinetic and metabolomic analyses to identify dysregulated human metabolic pathways. Then, AP arms will follow a similar analysis framework with an exploration of the additional dose-dependent effects. This study utilises a crossover design to reduce the intra-subject variability to compare the metabolomics of the metformin and AP formulations. Both medicines exhibit antidiabetic effects from the literature, so the human metabolic pathways related to antihyperglycemic effects using pharmacometabolomics will be explored.

### Objectives

The study aimed to determine the safety of metformin 1000 mg, AP capsules 1000 mg, and AP capsules 2000 mg in the clinical part.

In the pharmacokinetics and metabolomics analyses, metformin 1000 mg will be used as a model drug for the determination of the pharmacokinetic parameters including the Cmax, Tmax, half-life (T1/2), and AUC0-infinity. The individualised Cmax of metformin will be used for the metabolomics analyses to identify dysregulated human metabolic pathways that are associated with diabetic effects.

The AP 1000 mg and AP 2000 mg analyses will follow the pharmacokinetics and metabolomics framework from the metformin arm in the second analysis. The pharmacokinetics for AP 1000 mg and AP 2000 mg will be compared for the three bioactive compounds, andrographolide, neoandrographolide, and deoxyandrographolide. The dysregulated human metabolic pathways identified from the metabolomics analyses for two doses will be compared based on the dose-dependent effects. 

Finally, the dysregulated human metabolic pathways associated with the diabetic effects will be compared for all three intervention arms.

## 2. Materials and Methods

### 2.1. Study Design

This was a single-centre, open-labelled, three periods, six sequences, crossover, single-dose oral administration, randomised-controlled trial in healthy volunteers under the fasting condition. Approximately 24 to 27 subjects were screened for a list of inclusion and exclusion criteria (Figure 1) to recruit up to 18 healthy volunteers with the estimation of 30–50% ineligibility. For AP formulation, the time to reach the maximum plasma concentration (Tmax) and the half-life (T_1/2_) were recorded at 1.63 to 2 h and 6 h, respectively [22,23]. In the package insert of metformin, the Tmax and T_1/2_ for metformin are 2 h and 6.2 h [24]. As a result, the wash-out period was determined to be seven days, which is more than five half-lives of the investigation products to ensure that the drug concentrations were below the lower limit of bioanalytical quantification in all subjects at the beginning of the next period [25]. Eligible subjects were randomised in a ratio of 1:1:1 into either metformin 1000 mg, AP capsules 1000 mg, or AP capsules 2000 mg. Figure 2 describes the SPIRIT schedule of enrolment, assessment, and intervention for the explanation to the trial subjects.

In the early-phase clinical trial, absorption, distribution, elimination, and its interaction and adverse reactions were studied in pharmacokinetics [26]. A bioavailability study measures the rate and extent of the active ingredients from the pharmaceutical dosage into the site of action, and provides information for the absorption, distribution, and elimination of a compound through a plasma concentration–time curve [27]. Generally, a crossover design is potentially affected by carry-over effects, sequence assignment, subject drops-out, and difficulties in analyses [28]. The crossover study was chosen in this single-dose oral administration proof-of-concept trial to reduce the number of subject recruitment and reduce the intrasubject variability in the metabolomics analyses. In addition, the relative short half-life of the investigational products also reduced the carry-over effects and shorten the trial period when compared to the parallel design.

The rationale of metformin selection as a model drug includes the well-known safety profiles of the first line antihyperglycemic drug and its inconclusive mechanism of action. There have been several metabolomics analyses for metformin on healthy volunteers and patients, and the perturbed human metabolic pathways associated with the pharmacodynamics effects remain inconsistent [29]. Therefore, this study focussed on the individualised pharmacokinetic guided Cmax samples to investigate the dysregulation of metabolic pathways under a controlled environment. The AP capsule is a traditional medicinal product registered under the Ministry of Health Malaysia for general well-being [30]. Several clinical trials conducted have demonstrated the effects on diabetes, ulcerative colitis, and upper-respiratory tract infections [19,31,32]. Most of the traditional herbal medicine could not provide evidence of the pharmacokinetics of multi-components in the drug development process. Kantae et al. proposed that the discovery of metabolite biomarkers using pharmacometabolomics could predict the pharmacokinetic and pharmacodynamic effects [33]. Therefore, this proof-of concept crossover study was designed to investigate individualised pharmacokinetic and pharmacometabolomic studies on metformin in the first analysis set as a model drug, followed by AP using a similar analysis framework in the second analysis set with additional dose-dependent effects. The dysregulated human metabolic pathways associated with the diabetic effects for the three investigational products were then compared.

### 2.2. Ethics Committee and Clinical Trial Registry

This study was reviewed and approved by the Medical Research Ethics Committee (MREC), Universiti Malaya Medical Centre. The ethics committee is recognised by the Forum for Ethical Review Committees in the Asian & Western Pacific Region [34]. The study approval letter (Appendix A), ethics committee composition and meeting attendance composition were in accordance with Good Clinical Practice. The research protocol version 4.0 dated 17 September 2019 (Appendix A), patient information sheet (Appendix A), and informed consent form (Appendix A) in both the English and Malay languages, trial advertisement, trial insurance, principal investigator’s curriculum vitae and Good Clinical Practice certificate, and case report form were approved by the Medical Research Ethics Committee, University Malaya Medical Centre (MREC ID Number 2018112-6848) on 18 September 2019. Protocol amendments, protocol deviations, and interim study reports were submitted to the ethical committee.

Independent safety data or efficacy data monitoring committee was formed in this crossover trial to determine the continuity of the study as the doses used for the investigational products are marketed products approved by the Ministry of Health Malaysia. However, serious adverse events or protocol deviations will be notified to the ethics committee according to the written procedure.

The study was also registered with the National Medical Research Registry, Ministry of Health Malaysia (registered number NMRR-18-2907-41284) and Clinicaltrials.gov (identifier number NCT04161404) [35]. The study was conducted in accordance with the Malaysian Guidelines for Good Clinical Practice [36] and the Declaration of Helsinki [37].

### 2.3. Conduct of Study

#### 2.3.1. Trial Site

The study was conducted at the Clinical Investigation Centre, University Malaya Medical Centre. The clinical research ward is equipped with ten hospital beds and an emergency trolley. This is a tertiary university hospital that has an emergency department, and the research ward is connected to an intensive care unit if an emergency occurs. The study personnel were trained for the study protocols and the trial’s detailed procedures. The training record was maintained at the site. The principal investigator delegated trial related activities to the study team and recorded the individual personnel’s responsibilities on the job delegation sheet. The trial activities schedule is presented in Appendix A for an explanation of the activities to the trial site personnel.

#### 2.3.2. Inclusion and Exclusion Criteria

The subject inclusion criteria comprised male subjects aged 18 to 45 years with a body mass index between 18.5 and 29.5 kg/m^2^, non-smokers who were legible, and willing to provide written informed consent. Subject exclusion criteria included subjects suffering from any chronic illness, history of a pre-existing bleeding disorder, alcohol or drug abuse, kidney or liver dysfunction, suffering any psychiatric disorder, positive HIV, or hepatitis B or C, clinically significant abnormal electrocardiogram or abnormal laboratory screening evaluation, systolic blood pressure of 100–139 mmHg, diastolic blood pressure of 60–89 mmHg, pulse rate of 60–100/min, oral temperature more than 37.5 °C, history of allergy to the investigated product, the existence of any surgical or medical condition under the principal investigator judgement’s that might interfere with the absorption, distribution, metabolism, or excretion of the drug or is likely to compromise the safety of the volunteers and the inability to communicate or cooperate.

#### 2.3.3. Informed Consent and Patient Withdrawal Criteria

Volunteers will be freely given a copy of the patient information sheet and informed consent form, either in the English or Malay language, prior to signing the informed consent form. The study team will explain the study details and inform the subjects that they could withdraw from the study at any time without giving any reason. In addition, subjects could discontinue from the study if they suffered from significant illness or violation of the protocol. The patient information sheet included a subject confidentiality clause to protect confidentiality before, during, and after the trial. Only the ethics committee and regulatory authority could access the trial-related data and they were always required to respect confidentiality. In addition, the results of the completed study were published without the disclosure of the subject’s identity. A copy of the signed informed consent form was given to the subject.

#### 2.3.4. Assignment of Intervention, Randomisation, and Blinding

The three-period, six-sequence, permuted block randomisation sequence will be generated using statistical software SAS Enterprise Guide version 7.1, SAS Institute Inc., Cary, NC, USA. The seed code 736522 was generated for the purpose of reproducibility, and the source code is attached in Appendix A. The randomisation code will be given to the pharmacist to perform the investigated product dispensing for three formulations: code A for metformin 1000 mg, code B for AP capsules 1000 mg, and code C for AP capsules 2000 mg.

Both the metformin tablets and AP capsules are registered with the Ministry of Health Malaysia. The investigational product accountability was recorded by the pharmacist. Dispensing activities were performed according to the line clearance procedures to avoid cross contamination or dispensing errors (Appendix A). Another pharmacist will monitor each dispensing activity and a duplicate sticker labelled with the investigational product name (Appendix A), protocol number, and other details will be attached to the envelope.

This will be an open-label pharmacokinetics and metabolomics study. Prior knowledge of the investigational product administration to the subject does not affect the bioavailability and metabolic effects of the drug. The blinding procedure will not be applied in this study so the investigator and the subjects are aware of the type of investigational products being administered during the clinical trials.

#### 2.3.5. Investigational Products Intervention

The subjects will be divided into two stations during the administration procedure of the investigational products (Appendix A). Subjects are either given metformin 1000 mg tablets, AP 1000 mg, or AP 2000 mg at each period according to the randomisation schedule (Figure 3). Subjects will be instructed to swallow the study drug without chewing with 240 mL of plain water at room temperature in the presence of the principal investigator. The label from the drug envelope will be stuck directly to the case report form and the actual time will be recorded immediately after the mouth is checked to ensure that all of the drugs have been administered. Subjects will not be allowed to drink water one hour before and two hours after the study drug administration. After dosing, subjects will be required to remain in a sitting or semi-reclining position for at least 4 h; subjects will also be restricted from taking part in any stressful physical activity.

#### 2.3.6. Sample Size Calculation

This is a proof-of-concept pharmacokinetics and pharmacometabolomics exploration study to determine the pharmacokinetic parameters and human metabolic pathways. The objective of a pharmacokinetics study in a phase I trial is to mainly measure the characteristic of the drug, so there is no statistical basis to determine the sample size. The number of subjects usually ranges from 4 to 24 subjects [38]. The United States guidance recommended a minimum of 12 subjects at each arm in the pharmacokinetics bioequivalence study [39]. Another sample size estimation approach for metabolomic analysis without the experimental pilot data proposed 12 subjects in each group using dynamic probabilistic principal component analysis or 18 subjects in each group for the probabilistic principal components and covariates analysis [40]. This study will recruit 18 subjects in each formulation considering the phase one features and analyses to plan for the pharmacokinetics and metabolomics.

#### 2.3.7. Screening and Data Collection

Subjects will be screened for the eligibility criteria according to the screening case report form after signing the informed consent form. The principal investigator will perform the physical examination, review the demographic data, electrocardiogram, vital signs, biochemistry, haematological, serological, urinalysis test, and perform the interview questionnaire to determine the subject’s eligibility. Eligible subjects will attend the dosing day at the trial site and will be asked to fast for at least ten hours to reduce the variability of drug absorption. Eligible subjects will then be asked to attend a method validation visit to collect additional blood and urine samples.

On the dosing day, the study coordinator will record the subject attendance list, subject enrolment list, subject identification list, and subject randomisation sheet. Subjects will be assigned to the randomisation group by the study coordinator. Pre-dose urine samples and plasma samples will be collected before dosing. After single-dose oral administration, fourteen blood samples will be collected at 0.5, 1, 1.5, 2, 2.5, 3, 3.5, 4, 5, 6, 8, 10, 12, and 24 h post-dose. The actual sampling time will refer to the subject plasma sampling schedule (Appendix A) and subject urine sampling schedule. An indwelling intravenous cannula will be used to collect the pre-dose and up to 12 h post-dose blood sample by syringe. The blood samples will then transferred into a pre-labelled potassium ethylenediaminetetraacetic acid (KEDTA) tube. The heparin-lock technique will be performed to prevent clotting of the blood by injecting 1 mL of heparinised saline in the indwelling cannula. Ambulatory blood will be collected at 24 h post-dose. The blood samples will then be centrifuged at 35,000 rpm for 15 min at 20 °C, and the plasma separated into three aliquots in cryotubes and stored at −80 °C in a freezer. The post-dose urine samples will be collected at three periods: period U1 (0–4 h post-dose), U2 (4–8 h post-dose), and U3 (8–12 h post-dose. The urine samples will then be separated into three aliquots in cryotubes and stored at −80 °C in a freezer. The study activities will be recorded on the case report form.

Food intake will be standardized throughout ward stay. Subjects will be provided with lunch at 12.20 pm, a tea break at 4.30 pm, and dinner at 7 pm. The calorie count for the main meal will be maintained at approximately 600 kcal and the tea break calories at around 350 kcal. Meal intake start time and end time will be recorded on the meal monitoring form.

The biological samples will be segregated according to the individual subjects for all of the sampling time points and processing batches after the study is completed. The details of the plasma segregation activities will follow the procedures according to the plasma and urine sample segregation forms.

### 2.4. Safety Assessment

The principal investigator will stay on-site to monitor any adverse events during the ward stay. Vital signs will be monitored every two hours at 10 a.m., 12 p.m. and 2 p.m. Any complaint from the subjects will be directly recorded into the case report form for adverse event classification. An adverse event that occurs will be classified according to the clinical safety data management guidance [41].

Stopping rules and the continual reassessment method should be applied to phase one dose-escalation trials [42,43,44]. However, the stopping rules will not be implemented in this study for the following reasons: the investigated products are registered medicines, and the administered dose will be according to the regulatory approved dose. The study will involve antidiabetic drugs taken under the fasting condition, and lunch will be given four hours post-dose. Although metformin does not pose a hypoglycaemia risk, additional random plasma glucose monitoring will be performed every half an hour up to four hours to monitor the hypoglycaemia effect [45]. In the case a clinically significant hypoglycaemia event is determined by the investigator, 20 g of glucose water will be given to the subject at any point in time.

Physical examinations will be performed on all subjects before being discharged from the ward. At the end of the last dosing period, the biochemistry, haematology, and urinalysis will be performed to ensure the subjects’ safety before being discharged. A post-study phone call follow-up will be made seven days after the last dose of the last visit for adverse event monitoring.

## 3. Data Collection, Management, and Statistical Analyses

### 3.1. Clinical Data

The clinical part will be mainly recorded in the case report form and the clinical laboratory results. After the study is completed, the principal investigator will sign off on each case report form and the data lock date will be confirmed. After the data lock is set, the data will be transcribed into a data management Excel sheet to monitor the data entry activity. The data management sheets will include the demographic data, vital signs, clinical laboratory data, actual sampling time points, dosing time points, and adverse events.

The forms and templates that will be used throughout the data collection, management, and statistical analyses are outlined in Figure 4. The subject demographic data and laboratory characteristics for screening and post-dose monitoring will be tabulated. The number of adverse drug reactions will also be recorded accordingly. A list of tables, listings, and figures will be generated for the study report (Appendix A).

### 3.2. Pharmacokinetics Analysis

This study will involve the collection of biological samples in triplicate. The first set of samples will be used for the pharmacokinetics analysis. The blood samples for each time point will be analysed using liquid chromatography mass spectrometry (LCMS) to determine the subject’s plasma concentration of the active ingredients: metformin, andrographolide, neoandrographolide, and deoxyandrographolide. Additional plasma samples will be collected for the purpose of method validation according to the guidelines for bioanalytical method validation [46]. The method validation parameters will include the accuracy, precision, calibration curve, and quality control samples during the study validation. The internal standard will be spiked into each sample to ensure consistency among all of the peaks.

The individual subject plasma concentration will be used for the pharmacokinetics profile calculation to produce the drug concentration time–curve. Non compartmental analysis will be performed using MATLAB SimBiology software. The pharmacokinetics profile of Cmax, AUC, Tmax, and volume of distribution (Vd) will be determined for metformin 1000 mg, AP 1000 mg, and 2000 mg.

### 3.3. Untargeted Metabolomics and Multivariate Data Analysis

In the metformin analyses, pharmacometabolomics analysis will be performed to explore the human metabolic pathways between the pre-dose and peak plasma concentration. The samples will be analysed using Agilent LCMS-QTOF for an untargeted metabolomics approach following the literature method with some modification [47]. The samples will be analysed using a reverse phase column with a guard column in both the positive and negative modes. The instrument method setting will follow the METLIN analysis methods [48]. The pool quality control samples will be used to condition the column and visualisation checking for the batch analysis.

The untargeted metabolomics workflow is presented in Appendix A. The chromatograms will be converted to the mzXML file using Global Natural Products Social (GNPS) package, University of California San Diego, California, United States [49]. MS spectral processing, normalisation, batch correction, multivariate analysis, and functional metabolic pathway analysis will be run on MetaboAnalyst web-based application (McGill University, Montreal, QC, Canada) to identify the human metabolic pathways [50]. In the multivariate analyses, principal component analysis (PCA) and partial least square discriminant analysis (PLSDA) will be used to inspect the region’s separation among the analysed time points. After this, a t-test will be used to identify the significantly differentiated compounds between the two-time point groups. The significant features will be processed in MS peaks to a pathway module to identify the dysregulated human metabolic pathways according to the Kyoto Encyclopaedia of Genes and Genomes (KEGG) database.

A similar analysis framework for metformin as described above will be repeated for AP 1000 mg and AP 2000 mg. There will be a head-to-head comparison of the pharmacokinetic profiles for both the AP arms. After this, the pharmacometabolomic identified dysregulated human metabolic pathways will be compared for the two doses in AP to investigate the dose-dependent effects. Finally, the dysregulated human metabolic pathways associated with the diabetic effects for the metformin, AP 1000 mg, and AP 2000 mg will be compared.

### 3.4. Reporting Framework

The study will be designed to use metformin as a model drug in the application of targeted peak plasma concentration in the metabolomics study. The clinical safety data, method validation, pharmacokinetics and metabolomics results of metformin will be reported in detail. This report suggests an application model for individualised pharmacokinetics and pharmacometabolomics study in an early-phase clinical trial in the drug development program. The results and the proposed pharmacometabolomic application framework in an early-phase clinical trial for the first analysis using metformin was recently published [51].

In the second report, the pharmacokinetics profiles for AP 1000 mg and AP 2000 mg will be presented. The results will include the dose-dependent effects of the dysregulated human metabolic pathways associated with the pharmacological effects. This report will suggest another application model for the pharmacokinetics and pharmacometabolomics study using multi-component herbal medicines in an early-phase clinical trial. Additional dose-dependent effects will be described in the report. Finally, the dysregulated human metabolic pathways associated with the diabetic effects will be compared for metformin and AP.

## 4. Discussion

This proof-of-concept study protocol aims to determine the pharmacokinetics and explore the metabolomics profiling of metformin and AP capsules for application in early-phase clinical trial. The protocol describes the conduct of clinical trials with comprehensive methods and Appendix A that comply with the principle of Good Clinical Practice and relevant trial-related forms to record the clinical outcomes.

The pharmacokinetics and bioavailability study could determine the rate and extent of the drug being absorbed into the human blood circulation [27]. From the first metformin analysis dataset, the individualised peak plasma concentration obtained from the pharmacokinetics study will be used to explore the human metabolic pathways. This approach can identify the dysregulated human metabolic pathways associated with the pharmacological effect accurately based on the peak plasma concentration of the drug. Therefore, the combination of a pharmacokinetics and pharmacometabolomics study could complement the interpretation of the human metabolic pathways during peak plasma concentration.

In the second phase analyses, the two doses of AP capsules arms provide additional information about the pharmacokinetics profiles characteristic of the three biomarkers, andrographolide, neoandrographolide, and deoxyandrographolide. The pharmacometabolomic analysis for the two doses of AP capsules potential identify dose-dependent metabolic signatures with intense metabolic features at affected human metabolic pathways associated with the pharmacodynamic effects.

This study will employ a crossover design instead of the typical 6 + 2 dose escalation first-in-human study design to reduce the variability of the trial subjects in metabolomic study, shorten the trial duration, and maximise the resources to recruit different subjects. The study protocol is practical for institutional academic research compared to complex industry sponsor regulatory trials while complying with the Good Clinical Practice principle. This study potential provides a framework for the application of a pharmacokinetics and metabolomics approach in a phase one trial, which could reveal valuable drug intervention dysregulated human metabolic pathways or molecular drug effects through metabolomics analyses.

## 5. Conclusions

The proof-of-concept study protocol combines a pharmacokinetics and pharmacometabolomics study using metformin as the model drug, followed by AP in two doses in an early-phase clinical trial. The study protocol describes a crossover randomised controlled trial in healthy volunteers under the fasting condition with details of the trial procedures, data collection, data management, and statistical analysis. Two reports will be produced to describe the results of each analysis and propose an application model of the integration of pharmacokinetics and pharmacometabolomics in the drug development program. The incorporation of pharmacometabolomics in a pharmacokinetics study under a randomised controlled clinical trial will help to explore the metabolic pathways of drugs and herbal medicines in a drug development program.

## Figures and Tables

**Figure 1 jcm-11-03931-f001:**
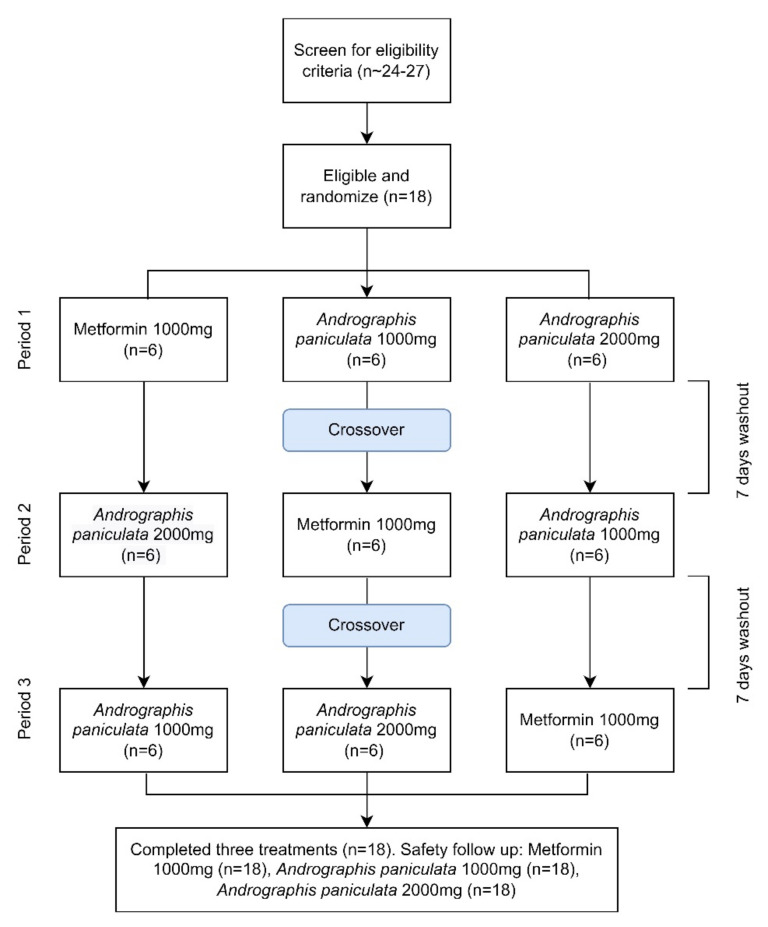
Crossover, three—treatments, three—period clinical trial study design.

**Figure 2 jcm-11-03931-f002:**
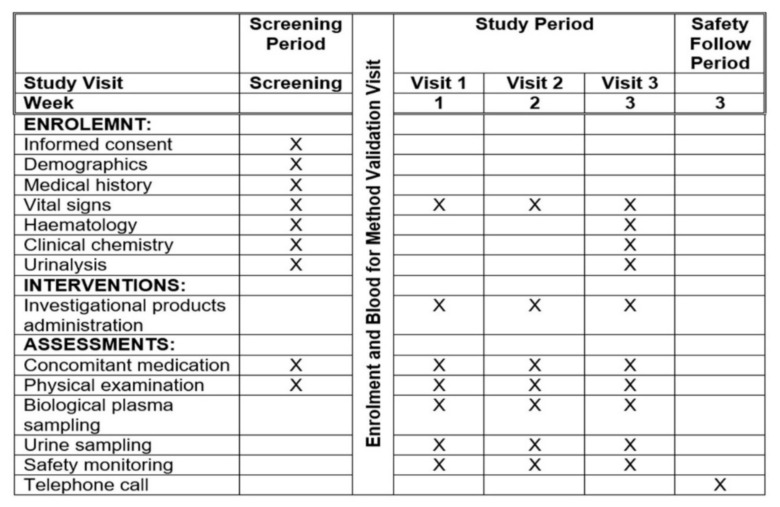
The SPIRIT schedule of enrolment, assessment, and interventions.

**Figure 3 jcm-11-03931-f003:**
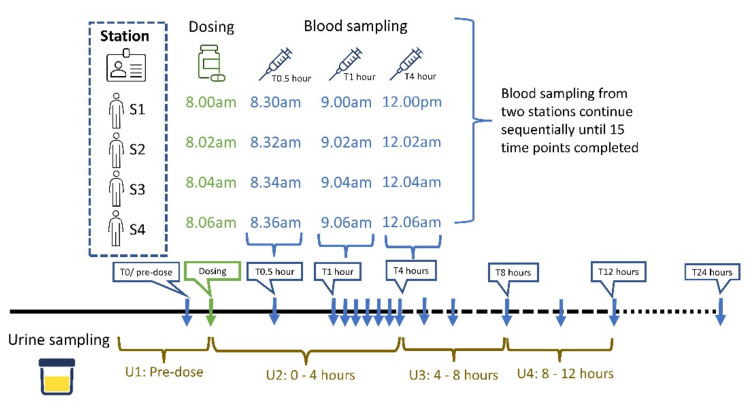
The diagram for dosing time, blood sampling, and urine sampling times in each period.

**Figure 4 jcm-11-03931-f004:**
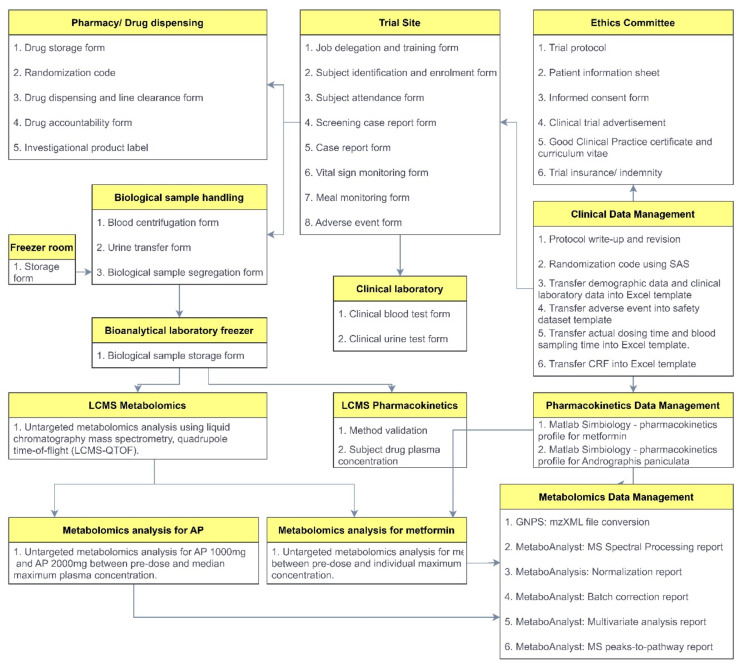
The forms and templates of the data collection, management, and statistical analyses.

## Data Availability

Not applicable.

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
