# Peer review of "Pharmacokinetics and Metabolomic Profiling of Metformin and Andrographis paniculata: A Protocol for a Crossover Randomised Controlled Trial"

_jcm, 2022, doi:10.3390/jcm11143931_

Round 1

Reviewer 1 Report

Title: Pharmacokinetics and metabolomic profiling of metformin and Andrographis paniculata: A protocol for a cross-over randomized controlled trial

Comment:

The study protocol by Tee et al., entitled "Pharmacokinetics and metabolomic profiling of metformin and Andrographis paniculata: A protocol for a cross-over randomized controlled trial" is intriguing research the authors used appropriate methods to get the results. The article is well-written and organized in general. However, some issues need to be addressed. I have given some comments for authors to consider while revising. The manuscript's language is frequently ambiguous and wordy; it must be improved. 

Comment 1: In the introductory section, it would be beneficial if the author could address the pharmacokinetics of Metformin and Andrographis Paniculata in general.

Comment 2: P8, L314: Data collection, management, and statistical analyses: It is suggested to write about the numerous forms of clinical study of the volunteers throughout data collecting and management.

Comment 3: Could you please provide a short detail about the Intervention model description

Comment 4: P2, L89 Authors are urged to expand on their descriptions of Andrographis Panicultata. Its origins, traditions, and certification for diabetic characteristics utilized in Malay

Comment 5: P8, L341 Author mentioned a single oral dose of metformin 1000mg. Just in tiny detail, discuss its Volume of distribution.

Comment 6:  Please double-check that there are no grammatical and punctuation issues in the manuscript.

Reviewer 2 Report

1.      This study is designed as a pharmacokinetic study but with additional intention to perform metabolomic analysis on the samples that that provide the Cmax/Tmax.

It is not clear what new information is being sought through this study. The pharmacokinetics of metformin is well known and well published. The pharmacokinetics of Andrographis paniculate has also been evaluated and published. (See - Wangboonskul et al. Pharmacokinetic study of Andrographis paniculata tablets in healthy Thai male volunteers. Thai Pharm Health Sci J. 2006;1(3):209-18. AND

Panossian et al. Pharmacokinetic and oral bioavailability of andrographolide from Andrographis paniculata fixed combination Kan Jang in rats and human. Phytomedicine. 2000 Oct 1;7(5):351-64.

2.      The study is not a phase one study and there is no clarity on why it is a proof-of-concept study.

3.      The need for a crossover design is not clear from the manuscript. Will there be a comparison between the PK data of metformin and Andrographis?

4.      Language – the many grammatical errors in the manuscript are a major distraction to any reader. The manuscript will benefit from extensive language editing.

Round 2

Reviewer 2 Report

Initial comments have been addressed.